# Application and Perceived Effectiveness of Complementary and Alternative Intervention Strategies for Attention-Deficit/Hyperactivity Disorder: Relationships with Affiliate Stigma

**DOI:** 10.3390/ijerph17051505

**Published:** 2020-02-26

**Authors:** Wen-Jiun Chou, Tai-Ling Liu, Ray C. Hsiao, Yu-Min Chen, Chih-Cheng Chang, Cheng-Fang Yen

**Affiliations:** 1College of Medicine, Chang Gung University, Taoyuan 33302, Taiwan; wjchouoe2@gmail.com; 2Department of Child and Adolescent Psychiatry, Chang Gung Memorial Hospital, Kaohsiung Medical Center, Kaohsiung 83301, Taiwan; 3Department of Psychiatry, Kaohsiung Medical University Hospital, Kaohsiung 80708, Taiwan; dai32155@gmail.com (T.-L.L.); bluepooh79@msn.com (Y.-M.C.); 4Department of Psychiatry, School of Medicine and Graduate Institute of Medicine, Kaohsiung Medical University, Kaohsiung 80708, Taiwan; 5Department of Psychiatry and Behavioral Sciences, University of Washington School of Medicine, Seattle, WA 98195-6560, USA; rhsiao@u.washington.edu; 6Department of Psychiatry, Children’s Hospital and Regional Medical Center, Seattle, WA 98105, USA; 7Department of Psychiatry, Chi Mei Medical Center, Tainan 70246, Taiwan; 8Department of Health Psychology, College of Health Sciences, Chang Jung Christian University, Tainan 71101, Taiwan

**Keywords:** attention-deficit/hyperactivity disorder, complementary and alternative intervention, caregivers, affiliate stigma

## Abstract

This cross-sectional questionnaire survey study was designed to examine the complementary and alternative intervention strategies (CAIS) employed by caregivers for their children’s attention-deficit/hyperactivity disorder (ADHD) and the associations of affiliate stigma with the employment and rated effectiveness of these strategies in Taiwan. A total of 400 caregivers of children with ADHD participated. CAIS that the caregivers employed and their effectiveness rated by the caregivers were surveyed. Associations of affiliate stigma with the application and rated effectiveness of the strategies were determined using logistic regression analysis. The results indicated that sensory integration (30.3%), exercise training (29.3%), sugar restriction (20.5%), and omega fatty acid supplementation (11.3%) were the most common CAIS that the caregivers employed. Caregivers with stronger affiliate stigma were more likely to employ sensory integration, exercise training, and omega fatty acid supplementation but also rated them as ineffective in treating their children’s ADHD. Various CAIS were employed by the caregivers to manage their children’s ADHD. Affiliate stigma was significantly associated with the application and rated ineffectiveness of several CAIS.

## 1. Introduction

### 1.1. Necessity of Treatment for Attention-Deficit/Hyperactivity Disorder

Attention-deficit/hyperactivity disorder (ADHD) is the most prevalent neuropsychiatric disorder worldwide [1]. A new study using a nationally representative sample of children in grades three, five, and seven reported a lifetime ADHD prevalence of 10.1% in Taiwan [2] according to the *Diagnostic and Statistical Manual of Mental Disorders*, Fifth Edition (DSM-5) diagnostic criteria [1]. People with ADHD have impaired adaptive functioning, manifested as a violation of rules, aggressive behavior, intolerance of gratification delay, unfocused attention, learning difficulties, impulsivity, and low motivation [3]. The core symptoms of ADHD may not only result in academic and occupational underachievement, interpersonal conflict, and strained familial relationships [3] but also increase the risk of suicide [4], traumatic brain injury [5], bone fracture [6], sexually transmitted infection [7], depression [8], and substance abuse [9]. Therefore, ADHD in children and adolescents warrants early diagnosis and intervention with effective strategies.

Pharmacologic management, such as with methylphenidate and atomoxetine, is the first-line treatment strategy for school-aged children with ADHD [10]. Regarding nonpharmacologic interventions, behavioral parent training is considered the first-line treatment in preschoolers with ADHD, though cognitive-behavioral therapy and child or parent training alone are not as effective as pharmacologic management for school-aged children [10,11].

### 1.2. Complementary and Alternative Intervention Strategies (CAIS) for ADHD

Many intervention options have been proposed for ADHD beyond the commonly used pharmacologic and cognitive-behavioral interventions [12]. It has long been known that use of complementary and alternative medicine was common among children who had received a diagnosis of ADHD or in whom ADHD was suspected [13]. According to the National Center for Complementary and Integrative Health, complementary interventions indicate nonmainstream practices used together with conventional medicine, while alternative interventions indicate nonmainstream practice used in place of conventional medicine [14]. A systematic review of studies on CAIS (cognitive training, neurofeedback, dietary omega fatty acid supplementation, herbal approaches, restriction and elimination diets, acupuncture, and homeopathy) concluded that little evidence exists across the outcome measures supporting the effectiveness of these strategies on treating ADHD [11]. Moreover, although evidence of the effectiveness of sensory integration for ADHD is limited and inconclusive [15], it is performed and promoted by physical therapists for the treatment of children’s ADHD in Taiwan [16]. Preference for only CAIS without evidence of their effectiveness may delay the use of effective treatment for children with ADHD, and untreated individuals have worse outcomes than treated patients in terms of academic, occupational, and social functioning [17].

### 1.3. Role of Affiliate Stigma for the Treatment of ADHD

According to ecological systems theory [18], preference for intervention models among caregivers of children with ADHD might be the result of interactions between individuals and environments. Caregivers’ beliefs and attitudes toward ADHD may form or change in the processes of interaction with families, peers, the media, and the societies they live in and may influence their search for assistance in managing their children’s symptoms and related problems [19]. Research found that higher levels of ADHD misconceptions were associated with lower acceptance of medication and higher acceptance of dietary interventions [20]. Research also found that parents seeing ADHD behaviors as more dispositional in children’s nature appears to be associated with exploration of nontraditional treatment alternatives [21].

The present study focused on the role of affiliate stigma in caregivers’ application and evaluation of the effectiveness of CAIS for ADHD. Affiliate stigma in caregivers of individuals with mental illness indicates that caregivers may perceive, be aware of, and internalize public stigma toward mental illness and their affiliates [22]. Affiliate stigma can result in affective, cognitive, and behavioral consequences for caregivers. For example, Mak and Cheung (2008) found that affiliate stigma among caregivers of children with mental disorders was positively associated with caregiver feelings of stress and subjective burden in Hong Kong [22]. A study in the United States indicated that heightened affiliate stigma in parents of children with ADHD is associated with more negative parenting and poorer social skills and increased aggression in children [23]. A study in France found that affiliate stigma in mothers of children with ADHD is positively related to both mothers’ distress and children’s symptoms [24]. However, neither the roles of affiliate stigma in the application of CAIS nor their subjective effects have been examined. If affiliate stigma is positively associated with the employment of CAIS for children’s ADHD, affiliate stigma warrants early intervention to provide timely assistance for caregivers and their children.

### 1.4. Aims of the Present Study

Similar to the people of other East Asian countries, Taiwanese people are deeply influenced by Confucianism and thus highly value children’s academic achievement, are collectivistic-oriented, and emphasize interpersonal relationships and harmony [25]. This sociocultural background may correspond to a lower tolerance among Taiwanese people to children’s uncooperativeness with rules and academic failure for any reason, including ADHD. People may blame family caregivers of children with ADHD for “not fulfilling their duty.” Therefore, family caregivers of children with ADHD may feel that they are “losing face” [26] and develop affiliate stigma. Moreover, family caregivers with affiliate stigma may deny a diagnosis of ADHD and the necessity of treatments that have been demonstrated to ameliorate the symptoms of ADHD. In this context, the present study examined the association of affiliate stigma with the employment of CAIS by family caregivers of children with ADHD in Taiwan and the effectiveness of these strategies as rated by caregivers. We hypothesized that affiliate stigma was positively associated with the employment of CAIS among caregivers and that affiliate stigma was negatively associated with the effectiveness of these strategies as rated by family caregivers.

## 2. Methods

### 2.1. Participants and Procedure

Main family caregivers of children 18 years of age or younger diagnosed with ADHD according to the diagnostic criteria in the DSM-5 [1] were recruited for this study between June 2018 and April 2019 from the child and adolescent psychiatric outpatient clinics of two medical centers in Kaohsiung, Taiwan. Two child psychiatrists conducted diagnostic interviews with the children and caregivers and made the ADHD diagnosis based on DSM-5 criteria. Multiple data sources—including clinical observation of each child’s behavior and caregivers’ ratings of ADHD symptoms on the short version of the Chinese version of the Swanson, Nolan, and Pelham, Version IV scale (SNAP-IV) [27,28] supported the diagnoses. Children who had an intellectual disability or autism spectrum disorder with communication difficulty were excluded. Main family caregivers meant family caregivers who spent the most time on caring for the children with ADHD compared with other caregivers. Main caregivers who had an intellectual disability, schizophrenia, bipolar disorder, or any cognitive deficits that resulted in significant difficulties in communication were also excluded. A total of 409 caregivers of children with ADHD were invited to participate in the study. Of these, nine (2.2%) declined to participate, leaving 400 (97.8%) caregiver participants (Figure 1). Regarding the sample size, a previous study found that 12% of children with a diagnosis of ADHD have received complementary and alternative medicine [13]. The sample of 400 participants was determined as adequate based on the estimation with 80% power, 95% confidence interval (CI), and statistically significant level (α) at 5% [29]. A total of 341 (85.3%) children received medication currently for their ADHD symptoms, and 59 (14.7%) received group cognitive-behavioral psychotherapy only currently. All main caregivers and their children with ADHD received psychoeducation about the etiology, symptoms presentation, and treatment strategies of ADHD. Main caregivers also received counseling about the skills to manage children’s ADHD symptoms and how to communicate with children. Most of the children without current medication treatment for ADHD have ever received medication before but stopped taking them because of intolerance to adverse effects. All participants provided written informed consent. The institutional review boards (IRBs) of Kaohsiung Medical University (KMUHIRB-E(I)-20180179) and Chang Gung Memorial Hospital Kaohsiung Medical Center (201800723A3) approved this study.

### 2.2. Measures

#### 2.2.1. CAIS for ADHD

First, we conducted three focus groups: Two groups for family caregivers of children with ADHD and one group for child psychiatrists to collect information regarding CAIS for ADHD. The family caregivers recruited into the focus groups have visited the child and adolescent psychiatric clinics for their children’s ADHD symptoms for at least two years. All participants had clear knowledge about standard treatment and CAIS for ADHD and were willing to share their experiences in the focus groups. Each group had five to eight members. The principal investigator (PI, CFY) led the focus groups by introducing the purpose of the focus groups and inviting the members to propose CAIS that family caregivers had employed and that caregivers and child psychiatrists had heard of. The PI also invited the members to clarify ambiguity and enhance discussion. Each focus group lasted for 40–50 min.

On the basis of the information collected, we developed a self-report questionnaire to assess the experiences of employing CAIS. The questionnaire first introduced the definitions of CAIS and then inquired whether family caregivers of children with ADHD had ever employed the following 15 CAIS for managing their children’s ADHD: Sensory integration, exercise training, sugar restriction, omega fatty acid supplementation, allergy treatment, music therapy, Chinese herbal medicine, meditation or mindfulness, chiropractic, acupuncture, homeopathy, folk therapy, mind growth programs of religious groups, neurofeedback, and use of a chelating agent for removal of heavy metals. If a caregiver answered yes to an item, then they were asked how old the child was when the caregiver employed the CAIS. Caregivers were also asked to rate on a 4-point scale to evaluate how effective they considered each CAIS was in improving their child’s ADHD symptoms, with 1 indicating “not effective at all,” 2 indicating “mildly effective,” 3 indicating “moderately effective,” and 4 indicating “very effective.” Since the distributions of responses on the items for effectiveness were skewed, those who were assigned a model score of 3 or 4 were classified as supporting its effectiveness for their children’s ADHD, and those assigning it a score of 1 or 2 were classified as not supporting its effectiveness.

#### 2.2.2. Caregivers’ and Children’s Factors

We used the affiliate stigma scale (ASS), a self-rated 22-item questionnaire, to evaluate the caregivers’ affiliate stigma. The ASS asks respondents to rate their agreement from 1 (strongly disagree) to 4 (strongly agree) using a 4-point scale. A higher score on the ASS indicates that the caregiver has a higher degree of affiliate stigma toward ADHD. The results of Kolmogorov–Smirnov and Shapiro–Wilk tests revealed that the scores of the ASS were normally distributed. The original version showed excellent internal consistency (α = 0.94) and satisfactory predictive validity [22]. The ASS was also confirmed to have valid and reliable psychometric properties in a Taiwanese sample [30]. Cronbach’s α was 0.95 in the present study.

The short version of the Chinese version of the SNAP-IV was used to assess the caregiver-reported severity of ADHD symptoms exhibited in the preceding month. This version comprises 26 items encompassing the core DSM-derived ADHD subscales of inattention, hyperactivity, and impulsivity, and the oppositional symptoms of oppositional defiant disorder [27,28]. Each item is rated on a 4-point scale ranging from 0 (not at all) to 3 (very much). A higher score on the subscales indicates a more severe inattention, hyperactivity/impulsivity, and oppositional symptoms. The Cronbach’s α in the present study for inattention, hyperactivity, and impulsivity, and oppositional behavior were 0.89, 0.90, and 0.92, respectively.

Caregivers’ and children’s sex, age, and education level were also collected. We also determined the caregivers’ marital status (married and living together vs. divorced or separated). The caregivers’ occupational socioeconomic status (SES) was assessed using the Close-Ended Questionnaire of the Occupational Survey (CEQ-OS) [31], which classifies paternal and maternal occupational SES into five levels, such that a high level indicates a high occupational SES. The CEQ-OS has been proven to have acceptable reliability and validity and has frequently been used in studies on children and adolescents in Taiwan [31]. For the purpose of statistical analysis in the present study, levels I, II, and III of the CEQ-OS were classified as low occupational SESs, and levels and IV and V were classified as high occupational SESs.

Finally, we asked caregivers how old the child was when they first visited a psychiatric clinical unit for children and adolescents and how effective they considered the treatments provided in the units to be in improving their child’s ADHD symptoms employing the scale used for rating the effectiveness of CAIS.

### 2.3. Procedure and Statistical Analysis

Before starting the study, the PI trained the research assistants to make sure that they were competent to direct the participants to complete the research questionnaire. Then, research assistants explained the procedures and methods of completing the questionnaire to the participants individually. The participants could propose any question when they had problems on completing the questionnaires, and the research assistants resolved their problems. The PI discussed with research assistants weekly to make sure of the quality of the study.

Data analysis was performed using the SPSS 22.0 statistical software (SPSS Inc., Chicago, IL, USA). Percentages for employing CAIS and their effectiveness were calculated. The age of the child when a CAIS was first employed was also included. To make sure of the statistical power, only the strategies employed by over 10% of the caregivers were selected into logistic regression analysis to examine the associations of affiliate stigma with the employment and effectiveness of the strategies. The caregiver’s sex, education level, marital status, and occupational SES and the child’s sex were covariates. Since most CAIS had been employed at the child’s younger age but not at the time of survey, the caregiver’s age and child’s age and current ADHD and oppositional symptoms were not included in the analysis. The p-value of wald χ^2^ and odds ratio (OR) and 95% confidence interval (CI) were used to indicate significance. A two-tailed *p-*value of less than 0.05 was considered statistically significant.

### 2.4. Ethics

The study procedures were performed in accordance with the Declaration of Helsinki. The IRBs of Kaohsiung Medical University Hospital and Chang Gung Memorial Hospital Kaohsiung Medical Center approved the study. All participants provided written informed consent before completing the questionnaires.

## 3. Results

### 3.1. Employment and Effectiveness of CAIS

Table 1 presents the caregiver and children demographic characteristics, affiliate stigma, and current ADHD and oppositional symptoms. The children’s scores of inattention and hyperactivity/impulsivity subscales on the SNAP-IV were 13.4 (3.6) and 9.8 (6.0), respectively. According to the norm of the Chinese Version of the SNAP- IV for ADHD in Taiwan [32], the scores indicated a mild severity. Table 2 presents the CAIS that caregivers employed to manage their child’s ADHD. The four CAIS that over 10% of the caregivers have employed were sensory integration (30.3%), exercise training (29.3%), sugar restriction (20.5%), and omega fatty acid supplementation (11.3%). Only 0.3%–8.8% of caregivers had employed any of other 11 strategies.

Sensory integration was employed at the youngest age (mean = 5.3 years, standard deviation (SD) = 2.3 years), followed by sugar restriction (mean = 6.0 years, SD = 2.3 years), exercise training (mean = 6.5 years, SD = 2.6 years), and omega fatty acid supplementation (mean = 7.1 years, SD = 3.5 years). Sensory integration, exercise training, and sugar restriction were rated by 70%–80% of caregivers as effective for ameliorating their child’s ADHD symptoms, whereas only 37.8% reported the effectiveness of omega fatty acid supplementation.

The mean age at first visit to a psychiatric clinical unit was 6.6 years (SD = 2.6 years). A total of 84.5% of caregivers rated the treatments provided by the clinical units, including medication and group cognitive-behavioral psychotherapy for their ADHD symptoms as effective.

### 3.2. Affiliate Stigma and Application of CAIS

Table 3 presents the results of logistic regression analysis on the association of affiliate stigma with the application of CAIS for treating ADHD. The p-value and OR indicated that after controlling for the effects of caregiver and child factors, affiliate stigma had weakly but significantly associations with the use of sensory integration, exercise training, and omega fatty acid supplementation.

### 3.3. Affiliate Stigma and Effectiveness of CAIS

Table 4 presents the results of logistic regression analysis on the association of affiliate stigma with the effectiveness of CAIS in ameliorating ADHD symptoms. The p-value and OR indicate that after controlling the effects of caregiver and child factors, affiliate stigma had weak but significant associations with the ineffectiveness of sensory integration, exercise training, and omega fatty acid supplementation, as rated by caregivers. No significant association was observed between affiliate stigma and ratings of the effectiveness of psychiatric treatment (OR = 0.978, 95% CI: 0.954–1.002).

### 3.4. Current Psychiatric Treatment, CAIS, and Affiliate Stigma

Differences in the application and effectiveness of CAIS and level of affiliate stigma were compared between groups of children who received medication currently for their ADHD symptoms (*n* = 341) and those who received only group cognitive-behavioral psychotherapy currently (*n* = 59). The results indicated that there were no differences in the use of sensory integration (*p* = 0.601), exercise training (*p* = 0.782), sugar restriction (*p* = 0.974), and omega fatty acid supplementation (*p* = 0.543) between the two groups. There were no differences in the rated effectiveness of sensory integration (*p* = 0.417), sugar restriction (*p* = 0.163), and omega fatty acid supplementation (*p* = 0.441) between the two groups. However, caregivers of children who received only group cognitive-behavioral psychotherapy currently were more likely to rate exercise training as ineffective than caregivers of children who received medication currently (*p* = 0.004). Moreover, caregivers of children who received only group cognitive-behavioral psychotherapy currently had a higher level of affiliate stigma (mean = 42.4; SD = 11.8) than did caregivers of children who received medication currently (mean = 38.1; SD = 11.1; *p* = 0.006).

## 4. Discussion

The present study is the first one to examine the relationship between affiliate stigma and the application and effectiveness of CAIS for ADHD among family caregivers of children with ADHD. Given that preference for only CAIS without evidence of their effectiveness may delay the use of effective treatment for children with ADHD, as well as that explanatory models of ADHD may influence caregivers’ decisions to choose intervention models for their children [33], the results of the present study indicated that health care professionals warrant routinely evaluating caregivers’ affiliate stigma when introducing and enhancing their motivation to receive evidence-based treatment models for their children.

We discovered that sensory integration, exercise training, sugar restriction, and omega fatty acid supplementation were the most common CAIS that family caregivers employed. Caregivers with a higher level of affiliate stigma were more likely to employ sensory integration, exercise training, and omega fatty acid supplementation but also more likely to rate them as ineffective for their child’s ADHD.

### 4.1. Affiliate Stigma and Employment of CAIS

Several etiologies may account for the relationship between affiliate stigma and application of CAIS. First, Mak and Cheung (2008) discovered a positive association between affiliate stigma among caregivers of children with mental disorders and caregivers’ feelings of stress and subjective burden [22]. Family caregivers may try multiple intervention strategies in addition to empirically supported treatment to manage their child’s ADHD symptoms and ameliorate their own stress and care burden. Second, given that affiliate stigma may result in not only negative affect (e.g., shame and embarrassment) and distorted cognition (e.g., worry and self-blame) but also behavioral consequences (e.g., avoiding social contact) [22], employing CAIS may be a behavioral consequence of affiliate stigma among family caregivers who avoid visiting medical units and facing the reality of their child’s ADHD. Third, parents of children with ADHD have lower social network supports than those of children without ADHD [34]. Moreover, family caregivers with stigma toward their child’s ADHD may socially isolate themselves from colleagues, friends, and family members [35,36]. Family caregivers with affiliate stigma may receive emotional and informational support from other caregivers who are also subject to affiliate stigma due to their child’s ADHD and then use CAIS introduced by these caregivers. Fourth, CAIS may not improve children’s ADHD. Persistent or even worsened symptoms may aggravate public prejudice against children with ADHD and their caregivers. Affiliate stigma in caregivers may become more serious. A vicious cycle of worsening ADHD symptoms and increased affiliate stigma forms. Although the cross-sectional design of the present study limited the possibility of determining a causal relationship between affiliate stigma and the application of CAIS, both delayed adoption of empirically supported treatments. Moreover, affiliate stigma may result in adverse consequences for children with ADHD and their caregivers. Therefore, health care professionals should routinely evaluate family caregivers’ options regarding treatment models and affiliate stigma.

### 4.2. Affiliate Stigma and the Effectiveness of CAIS

Treatments, including medication and group cognitive-behavioral psychotherapy provided in child and adolescent psychiatric clinical units were rated by 84.5% of family caregivers as effective in improving their child’s ADHD symptoms. However, 70%–80% of the caregivers rated sensory integration, exercise training, and sugar restriction as effective in ameliorating their children’s ADHD symptoms, whereas only 37.8% supported the effectiveness of omega fatty acid supplementation. The results suggest that caregivers of children with ADHD may rate certain CAIS as effective in ameliorating children’s ADHD symptoms even when such strategies lack adequate empirical support regarding their effectiveness. The present study neither surveyed details of CAIS (such as duration, intensity, modules, and quality of execution) nor used the standard rating scales for measuring their effectiveness. Family caregivers may have viewpoints and expectations not identical to treatment goals that health care professionals regard as priorities in treating ADHD. For example, a previous study found that parents of children with ADHD rated improvements in the child’s social situation and emotional state as the most important in treating ADHD and suggested that treatment decisions for children with ADHD should include parents’ preferences to improve clinical outcomes [37].

A higher level of affiliate stigma was significantly associated with the ineffectiveness of sensory integration, exercise training, and omega fatty acid supplementation as rated by caregivers, raising the possibility that family caregivers with affiliate stigma may also rate the treatment provided in psychiatric units as ineffective. However, no significant association was observed between affiliate stigma and ratings of the effectiveness of psychiatric treatment. How affiliate stigma influences family caregivers’ rating of the effectiveness of CAIS warrants further study.

### 4.3. Current Psychiatric Treatment, CAIS, and Affiliate Stigma

The present study did not find significant differences in the use and rated effectiveness of common CAIS between the children receiving medication and receiving group cognitive-behavioral psychotherapy currently. However, caregivers of children receiving psychotherapy were more likely to rate exercise training as ineffective than caregivers of children receiving medication currently. Caregivers of children receiving psychotherapy also had a higher level of affiliate stigma than did caregivers of children receiving medication currently. Tracing back the history, we found that most of the children receiving psychotherapy but no medication currently had ever received medication before but stopped taking medication because of intolerance to adverse effects. It raised the possibility that caregivers’ affiliate stigma may relate to children’s intolerance to adverse effects or caregivers’ evaluation for children’s adverse response to medication. It warrants further study.

### 4.4. Limitations

The present study had several limitations. First, as mentioned, its cross-sectional design limited the possibility of determining a causal relationship between affiliate stigma with employment and effectiveness of CAIS. Second, we retrospectively obtained data on employment and the effectiveness of strategies; therefore, recall bias might have been introduced. Third, we recruited caregivers from psychiatric outpatient clinics. The results might not be generalizable to family caregivers who have never brought their children with ADHD to a medical unit. Fourth, few family caregivers had ever employed CAIS other than sensory integration, exercise training, sugar restriction, and omega fatty acid supplementation. We could not determine the relationship between affiliate stigma and these other strategies. The results of logistic regression analysis examining the association between affiliate stigma and application and effectiveness of some CAIS revealed a large 95% CI, indicating a low level of precision of the OR [38]. Fifth, the children with ADHD in the present study were mainly boys (84%). A study using the National Health Insurance Research Database of Taiwan to explore trends in ADHD diagnosis from 2000 to 2011 among youths in Taiwan revealed that 21.4% of cases were girls and 78.6% were boys [39]. The small number of caregivers of girls in the present study may limit generalizability of findings to families of girls with ADHD. Although the present study measured children’s current ADHD symptoms, we did not determine the subgroups of children based on the DSM-5 inattentive, hyperactive, and combined presentations.

## 5. Conclusions

The present study found that affiliate stigma was significantly associated with the application and ratings of ineffectiveness for sensory integration, exercise training, and omega fatty acid supplementation by the family caregivers of children with ADHD. Based on the results of the present study, we proposed four suggestions. First, one-third of caregivers have ever employed sensory integration or exercise training, one-fifth employed sugar restriction, and over one-tenth employed omega fatty acid supplementation for their children’s ADHD. Interestingly, 70%–80% of caregivers rated sensory integration, exercise training, and sugar restriction as effective for ameliorating their child’s ADHD symptoms. How the caregivers evaluated the effectiveness of these CAIS warrants further study. Especially, it needs further study why the caregivers rated CAIS as effective but visited the psychiatric clinics for further evaluation and treatment. Caregivers’ subjective experiences should be emphasized and carefully examined to identify their needs [40]. Health care professionals should perform an in-depth investigation of family caregivers’ motivation for employing CAIS for their child’s ADHD rather than simply attributing it to caregivers’ ignorance or denial of illness.

Second, the significant association between affiliate stigma and the application of several CAIS for ADHD indicated the possibility that application of CAIS may be a strategy of caregivers to cope with their affiliate stigma. Given that preference of CAIS but not evidence-based treatment for ADHD may delay the timing of managing children’s ADHD symptoms, health care professionals should view affiliate stigma as a critical topic that warrants vigorous evaluation and intervention.

Third, although no intervention program has been proposed to effectively reduce caregivers’ affiliate stigma, intervention programs should be “ecologically sensitive” and view affiliate stigma as the results of interactions among multiple ecological systems. Children with ADHD and their caregivers may benefit from treatment in which children, caregivers, social environments, and broader political and cultural contexts that shape children’s behaviors are carefully investigated [41].

Fourth, although the present study did not examine whether the caregivers obtained the information of CAIS from the internet, research has demonstrated that the internet becomes the first and most popular source for caregivers to search for opinions regarding the etiologies and intervention models of ADHD [42]. In fact, there are many messages of CAIS for ADHD that can be detected in the internet. Health care professionals can use the internet to deliver knowledges about evidence-based etiologies and treatment models of ADHD to the internet users. In particular, the internet can be also used to educate people and mitigate distorted images spread in the media of ADHD, and it may have meaningful implications for reducing the development of affiliate stigma.

## Figures and Tables

**Figure 1 ijerph-17-01505-f001:**
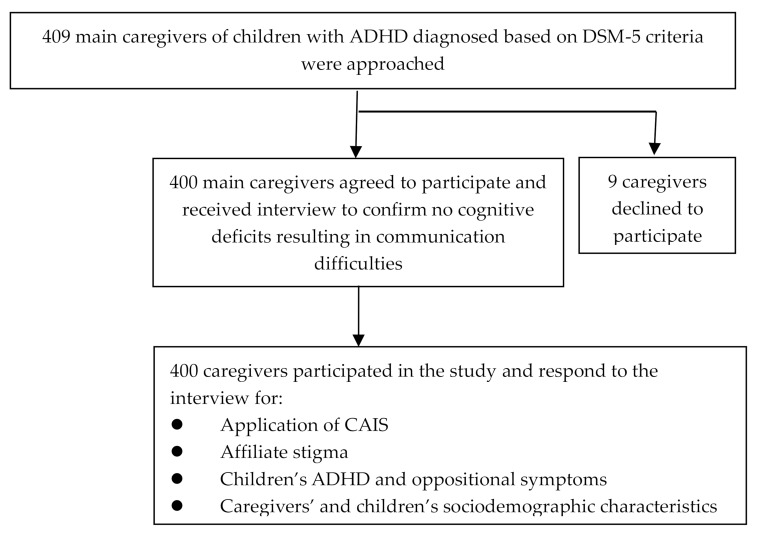
Flowchart of study design. ADHD: Attention-deficit/hyperactivity disorder; CAIS: Complementary and alternative intervention strategies; DSM-5: Diagnostic and Statistical Manual of Mental Disorders, Fifth Edition.

**Table 1 ijerph-17-01505-t001:** Caregivers’ demographic characteristics and affiliated stigma and children’s demographic characteristics and ADHD symptoms (N = 400).

Variables	*n* (%)	Mean (SD)	Range
***Caregivers***			
Relationship with the child			
Mother	287 (71.8)		
Father	90 (22.5)		
Others	23 (5.8)		
Age (years)		43.4 (6.8)	25–70
Sex			
Female	304 (76.0)		
Male	96 (24.0)		
Education (years)		13.8 (2.9)	3–23
Parental marriage status			
Intact	320 (80)		
Disruptive	80 (20)		
Occupational socioeconomic status			
High	155 (38.8)		
Low	245 (61.2)		
Affiliate stigma		38.7 (11.3)	22–75
***Children***			
Age (years)		10.7 (3.2)	4–18
Sex			
Girls	64 (16.0)		
Boys	336 (84.0)		
Education			
Primary school or kindergarten	355 (88.8)		
High school	45 (11.3)		
ADHD symptoms on the SNAP-IV			
Inattention		13.4 (3.6)	0–27
Hyperactivity/impulsivity		9.8 (6.0)	0–27
Opposition defiance		10.1 (6.0)	0–24

ADHD: Attention-deficit/hyperactivity disorder; SNAP-IV: Swanson, Nolan, and Pelham, Version IV Scale

**Table 2 ijerph-17-01505-t002:** CAIS employed by the caregivers for children’s ADHD (N = 400).

Variables	*n* (%)	Age of Children at the First Visit (Years) Mean (SD)	Effect	
			No Effect *n* (%)	Have Effect *n* (%)
Sensory integration	121 (30.3)	5.3 (2.3)	35 (28.9)	86 (71.1)
Exercise training	117 (29.3)	6.5 (2.6)	23 (19.7)	94 (80.3)
Sugar restriction	82 (20.5)	6.0 (2.3)	26 (31.7)	56 (68.3)
Omega fatty acids supplement	45 (11.3)	7.1 (3.5)	28 (62.2)	17 (37.8)
Treatment for allergy	35 (8.8)	6.7 (2.9)	19 (54.3)	16 (45.7)
Music therapy	32 (8.0)	6.5 (2.9)	10 (31.3)	22 (68.7)
Chinese herbal medicine	21 (5.3)	7.9 (3.3)	12 (57.1)	9 (42.9)
Meditation or mindfulness	10 (2.5)	7.2 (3.6)	1 (10)	9 (90)
Chiropractic	10 (2.5)	5.9 (3.1)	2 (20)	8 (80)
Acupuncture	9 (2.3)	7.0 (3.9)	4 (44.4)	5 (55.6)
Homeopathy	9 (2.3)	5.1 (3.1)	5 (55.6)	4 (44.4)
Folk therapy	7 (1.8)	5.4 (4.0)	2 (28.6)	5 (71.4)
Mind growth programs by religious groups	7 (1.8)	7.0 (4.3)	2 (28.6)	5 (71.4)
Neurofeedback	1 (0.3)	10 (n/a)	0	1 (100)
Chelating agent for removing heavy metals	1 (0.3)	3 (n/a)	1 (100)	0

**Table 3 ijerph-17-01505-t003:** Factors related to the employment of complementary and alternative intervention strategies (CAIS) for ADHD.

Variables	Sensory Integration		Exercise Training		Sugar Restriction		Omega Fatty Acids Supplement	
	OR (95% CI)	*p*	OR (95% CI)	*p*	OR (95% CI)	*p*	OR (95% CI)	*p*
***Caregivers***								
Males ^a^	0.405 (0.220–0.746)	0.004	0.703 (0.401–1.235)	0.221	0.440 (0.215–0.902)	0.025	0.432 (0.160–1.168)	0.098
Age	0.987 (0.953–1.023)	0.485	0.986 (0.952–1.021)	0.424	0.986 (0.944–1.029)	0.516	0.976 (0.923–1.031)	0.387
Education level	1.072 (0.985–1.167)	0.108	1.036 (0.952–1.126)	0.412	1.100 (0.998–1.211)	0.054	1.034 (0.914–1.170)	0.592
Disruptive marriage status ^b^	0.897 (0.512–1.574)	0.706	1.046 (0.603–1.815)	0.873	0.586 (0.289–1.187)	0.138	0.496 (0.187–1.317)	0.159
Low occupational SES ^c^	0.878 (0.537–1.436)	0.605	1.207 (0.734–1.984)	0.458	0.701 (0.404–1.218)	0.207	1.038 (0.506–2.132)	0.918
Affiliate stigma	1.020 (1.000–1.040)	0.049	1.023 (1.003–1.043)	0.022	1.015 (0.993–1.038)	0.184	1.042 (1.013–1.071)	0.004
***Children***								
Boys ^d^	1.405 (0.749–2.634)	0.290	0.926 (0.512–1.675)	0.799	0.790 (0.410–1.524)	0.483	1.700 (0.664–4.354)	0.269

SES: Socioeconomic status. ^a^ female as reference; ^b^ intact marriage as reference; ^c^ high socioeconomic status as reference; ^d^ girls as reference.

**Table 4 ijerph-17-01505-t004:** Factors related to the effectiveness of CAIS for ADHD evaluated by caregivers.

Variables	Sensory Integration		Exercise Training		Sugar Restriction		Omega Fatty Acids Supplement	
	OR (95% CI)	*p*	OR (95% CI)	*p*	OR (95% CI)	*p*	OR (95% CI)	*p*
***Caregivers***								
Males ^a^	1.077 (0.297–3.897)	0.911	0.520 (0.141–1.916)	0.326	1.304 (0.287–5.928)	0.731	0.105 (0.007–1.632)	0.107
Age	0.930 (0.864–1.002)	0.055	0.944 (0.876–1.017)	0.131	0.982 (0.911–1.058)	0.634	0.922 (0.789–1.079)	0.311
Education level	1.093 (0.925–1.291)	0.296	1.107 (0.919–1.332)	0.285	1.010 (0.843–1.210)	0.915	0.758 (0.516–1.114)	0.158
Disruptive marriage status ^b^	0.994 (0.300–3.288)	0.992	3.054 (0.608–15.336)	0.175	1.074 (0.243–4.744)	0.925	7.863 (0.260–237.923)	0.236
Low occupational SES ^c^	0.679 (0.262–1.761)	0.426	0.546 (0.153–1.946)	0.350	0.567 (0.199–1.617)	0.288	0.690 (0.1144–0.169)	0.686
Affiliate stigma	0.945 (0.908–0.984)	0.006	0.955 (0.913–0.999)	0.043	0.976 (0.936–1.018)	0.261	0.909 (0.844–0.980)	0.013
***Children***								
Boys ^d^	0.170 (0.031–0.939)	0.042	1.829 (0.526–6.358)	0.342	1.356 (0.401–4.586)	0.624	0.284 (0.023–3.527)	0.327

SES: Socioeconomic status. ^a^ female as reference; ^b^ intact marriage as reference; ^c^ high socioeconomic status as reference; ^d^ girls as reference.

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
