# Peer review of "Application and Perceived Effectiveness of Complementary and Alternative Intervention Strategies for Attention-Deficit/Hyperactivity Disorder: Relationships with Affiliate Stigma"

_ijerph, 2020, doi:10.3390/ijerph17051505_

Round 1

Reviewer 1 Report

This study aims to examine the influence of affiliate stigma on the use of alternative intervention strategies employed by caregivers for their children’s attention-deficit/hyperactivity. It was carried out on Taiwan a society characteristics associated to Confucianism.

The study is well organized and done but their conclusions have to be understood within the framework of this society.

Nevertheless, there are some important points to be taken into account in order further understand and to have a more holistic and whole idea about the topic assessed in this study:

The percentage of people using the standard interventions strategies for ADHD To assess the level of affiliate stigma among people using standard interventions for ADHD.

Author Response

We appreciate your comments on our manuscript. As discussed below, we have revised our manuscript with underlines according to the reviewers. The following responses have been prepared to address your comments in a point-by-point fashion. Please let us know if there is anything else we should provide.

Comment

The study is well organized and done but their conclusions have to be understood within the framework of this society.

Response

Thank you for your comment. In the revised manuscript we extended the contents of Conclusion to propose suggestions about how to reduce affiliate stigma and increase knowledge of evidence-based treatment for ADHD in caregivers based on the results of the present study. Please refer to line 357-386. We also added the studies on affiliate stigma among caregivers of children with ADHD in Hong Kong, the United States and France to demonstrate that affiliate stigma of ADHD is a worldwide issue warranted evaluation. Please refer to line 92 and 94.

Comment

Nevertheless, there are some important points to be taken into account in order further understand and to have a more holistic and whole idea about the topic assessed in this study.

Response

Thank you for your comment. As mentioned above, we rewrote the contents of Conclusion to propose suggestions, especially the importance of intervention programs with ecological sensitivity. Please refer to line 357-386.

Comment

The percentage of people using the standard interventions strategies for ADHD To assess the level of affiliate stigma among people using standard interventions for ADHD.

Response

A total of 341 (85.3%) and 59 (17%) children received medication and group cognitive-behavioral psychotherapy currently for their ADHD symptoms. All caregivers and their children with ADHD received psychoeducation about the etiology, symptoms presentation, and treatment strategies of ADHD. Caregivers also received counseling about the skills to manage children’s ADHD symptoms and how to communicate with children. Most of children without current medication treatment for ADHD have ever received medication before but stopped taking because of intolerance to adverse effects. We added the descriptions above into the revised manuscript. Please refer to line 134-140. Given that nearly all children currently received or have ever received medication for ADHD symptoms, we did not compare the level of affiliate stigma between caregivers of children with and without medication for ADHD,

Reviewer 2 Report

The Authors planned the study to examine the alternative intervention strategies employed by caregivers for their children's ADHD and the associations of affiliate stigma with employment and rated effectiveness of these strategies in Taiwan.

Multiple and ambitious purposes common to all caregivers of mental health patients of all ages. Much has been produced in this regard, but much still needs to be done in particular on how and what to modify in practice to reduce stigma, increase effectiveness of interventions.

At the end of the reading the answer to the question of whether this article contributes in this direction is no. It is not clear what the present work adds again and what to do to improve (perspectives). Authors should make explicit these points.

"alternative" should be better defined compared to the prescribed, followed, interrupted therapy, as well as the documented (RCTs, systematic reviews, guidelines) evidence of effectiveness.

"alternative" it does not necessarily mean only this can be added.

The word "compliance" never appears in the text yet it is closely linked to "perceived".

The sample size it is too small to properly analyze the numerous variables associated with population characteristics, many of which are ignored.

According to the criteria of selection it is not clear who the stakeholders involved are representative of.

The wide SD and CI, as OR value (none really significant) do not support conclusions.

Perceived vs documented effectiveness of different interventions should be discussed.

In the present form the manuscript it is not very linear and questionable in the methodological approach. Authors should try to work in this direction to improve it.

Author Response

revised our manuscript with underlines according to the reviewers. The following responses have been prepared to address your comments in a point-by-point fashion. Please let us know if there is anything else we should provide.

Comment

Multiple and ambitious purposes common to all caregivers of mental health patients of all ages. Much has been produced in this regard, but much still needs to be done in particular on how and what to modify in practice to reduce stigma, increase effectiveness of interventions.

Response

Thank you for your comment. In the revised manuscript we extended the contents of Conclusion to propose suggestions about how to reduce affiliate stigma and increase knowledge of evidence-based treatment for ADHD in caregivers based on the results of the present study. Please refer to line 357-386.

Comment

At the end of the reading the answer to the question of whether this article contributes in this direction is no. It is not clear what the present work adds again and what to do to improve (perspectives). Authors should make explicit these points.
Response

We added explanation for the possible contribution of this study to the field of ADHD study as below. Please refer to line 277-284.

“The present study is the first one to examine the relationship between affiliate stigma and the application and effectiveness of CAIS for ADHD among family caregivers of children with ADHD. Given that preference for only CAIS without evidence of their effectiveness may delay the use of effective treatment for children with ADHD, as well as that explanatory models of ADHD may influence caregivers’ decisions about what type of interventions to pursue for their children [33], the results of the present study indicated that health care professionals warrant routinely evaluating caregivers’ affiliate stigma when introducing and enhancing their motivation to receive evidence-based treatment models for their children.”

As mentioned above, we extended the contents of Conclusion to propose suggestions about how to reduce affiliate stigma and increase knowledge of evidence-based treatment for ADHD in caregivers based on the results of the present study. Please refer to line 357-386.

Comment

"alternative" should be better defined compared to the prescribed, followed, interrupted therapy, as well as the documented (RCTs, systematic reviews, guidelines) evidence of effectiveness.

Response

Thank you for your suggestion. In the revised manuscript we added the definition of alternative intervention according to the National Center for Complementary and Integrative Health. We also included complementary intervention strategies into the present study to comprise the intervention strategies examined in the present study. Please refer to line 63-66. We defined the standard treatment strategies including pharmacologic management for school-aged children with ADHD and parent behavioral training for preschoolers with ADHD (Charach et al., 2011; Goode et al., 2018). Please refer to line 54-58. Other intervention strategies are identified as complementary and alternative intervention strategies (CAIS).

Comment

"alternative" it does not necessarily mean only this can be added.

Response

Thank you for your reminding. We revised alternative intervention strategies into complementary and alternative intervention strategies (CAIS) to comprise the intervention strategies examined in the present study. Please refer to the topic and thorough the revised manuscript.

Comment

The word "compliance" never appears in the text yet it is closely linked to "perceived". 
Response

In the original manuscript we have mentioned:

”The present study neither surveyed details of CAIS (such as duration, intensity, modules, and quality of execution) nor used the standard rating scales for measuring their effectiveness. Family caregivers may have viewpoints and expectations not identical to those that health care professionals regard as priorities in treating ADHD. For example, a previous study found that parents of children with ADHD rated improvements in the child's social situation and emotional state as the most important in treating ADHD and suggested that treatment decisions for children with ADHD should include parents' preferences to improve clinical outcomes.” Please refer to line 322-329.

Comment

The sample size it is too small to properly analyze the numerous variables associated with population characteristics, many of which are ignored.

Response

We added a paragraph to illustrate the sample size. Please refer to line 130-134.

“Regarding the sample size, a previous study found that 12% of children with a diagnosis of ADHD have received complementary and alternative medicine [13]. The estimated sample size was 386 with 80% power, 95% confidence interval (CI), and statistically significant level (α) at 5% [29]. The sample of 400 participants was thus determined as adequate.”

Comment

According to the criteria of selection it is not clear who the stakeholders involved are representative of.

Response

Main family caregivers of children with ADHD were invited into this study. Main family caregivers meant family caregivers who spent the most time on caring the children with ADHD compared with other caregivers. We added it in line 125-127.

Comment

The wide SD and CI, as OR value (none really significant) do not support conclusions.

Response

The main continuous variable in the present study is the total score of the Affiliate Stigma Scale (ASS). The results of Kolmogorov-Smirnov and Shapiro-Wilk tests revealed that the scores of the ASS were normally distributed. We added the explanation in line 180-181. To support the significance of statistical results, we added p value into the revised manuscript. Please refer to line 221-223 and Tables 3 and 4. The results of logistic regression analysis examining the association between affiliate stigma and application and effectiveness of some CAIS revealed a large 95% CI, indicating a low level of precision of the OR [38]. We added it as one of limitations of this study. Please refer to line 346-348. We also labelled the significant results as “weakly but significantly associations.” Please refer to line 267-268.

Comment

Perceived vs documented effectiveness of different interventions should be discussed.

Response

This issue was discussed as below. Please refer to line 320-329.

”The present study neither surveyed details of CAIS (such as duration, intensity, modules, and quality of execution) nor used the standard rating scales for measuring their effectiveness. Family caregivers may have viewpoints and expectations not identical to those that health care professionals regard as priorities in treating ADHD. For example, a previous study found that parents of children with ADHD rated improvements in the child's social situation and emotional state as the most important in treating ADHD and suggested that treatment decisions for children with ADHD should include parents' preferences to improve clinical outcomes.

Comment

In the present form the manuscript it is not very linear and questionable in the methodological approach. Authors should try to work in this direction to improve it.

Response

Thank you for your suggestion. In the revised manuscript we added a flowchart of study design to illustrate the methodological approach. Please refer to Figure 1 (line 146).

Reviewer 3 Report

This is a well written and interesting manuscript. I have a few comments to improve the clarity of the methods:

Regarding inclusion criteria, were all ADHD subtypes included (inattentive, hyperactive/impulsive, combined)? Would be useful to include in Table 1. Would also be interested in whether the results were impacted by subtype. In 2.1, the authors mention that the caregivers "were interviewed by research assistants by using the research questionnaire." Were there interviews in addition to questionnaires? If so, what was the reliability among the research assistants in the conduction of an interview? Apologies if I missed this, but what proportion of the sample received traditional ADHD treatments (i.e., medication, parent training) in the past or currently? Were any of the children still medicated and/or in parent training treatments, and if so, may need to consider in statistical models? More information about the focus group process would be useful to the reader. How were caregivers selected for participation for focus groups? Were they part of the larger 400 sample? What queries were used, who led the groups, how long did groups last, etc? In 2.2.1, the authors mention development of a questionnaire. It would be helpful to specify that the questionnaire was then used with the full sample of 400. Not all readers may be familiar with the SNAP-IV. Are there clinical cutoffs? Would be helpful to include these in the paper to ease interpretation of the means presented in Table 1. (i.e., how clinically severe are the children on average?) In 2.3, was there a specific rationale for selecting 10% as the cutoff (as opposed to a different cutoff) for including a given strategy in logistic regressions? In Table 1, it might be helpful to change the font of "caregivers" and "children" to italics, bold, or underlined to help the reader quickly orient to the categories of demographics included in the table. Tables would benefit from an indicator within the table (asterick, bolded text, or similar) to highlight which analyses are significant. The sample is heavily male - is this representative of the distribution of ADHD across the sexes in Taiwan? May limit generalizability of findings to families of girls with ADHD. Discussion - I wondered how perceptions of effectiveness of alternative interventions might compare to perceptions of effectiveness of traditional interventions (either in this study or more broadly in the literature). on Page 10, the authors mentioned that "treatments provided in child and adolescent psychiatric clinical units were rated by 84% of family caregivers as effective." Does this refer to traditional interventions, and was this in the current sample (I think so, but I didn't see it in the results)? Similarly, might suggest moving the Odds ratio reported in 4.2 to results section.

Author Response

We appreciate your comments on our manuscript. As discussed below, we have revised our manuscript with underlines according to the reviewers. The following responses have been prepared to address your comments in a point-by-point fashion. Please let us know if there is anything else we should provide.

Comment

Regarding inclusion criteria, were all ADHD subtypes included (inattentive, hyperactive/impulsive, combined)? Would be useful to include in Table 1. Would also be interested in whether the results were impacted by subtype.

Response

The present study recruited the caregivers of children with ADHD diagnosed based on DSM-5 The DSM-5 has removed the subtypes for ADHD. Therefore, we used the SNAP-IV to measure the severities of inattention and hyperactivity/impulsivity. Because of children recruited from the child and adolescent psychiatric clinics and receiving treatment, their scores of inattention and hyperactivity/impulsivity subscales on the SNAP-IV were 13.4 (3.6) and 9.8 (6.0), respectively, indicating mild levels of severities. We added the introduction of the SNAP-IV and the interpretation in the revised manuscript. Please refer to line 189-191 and line 232-234. Moreover, it is hard to determine the causal relationship between ADHD symptoms and application of complementary and alternative intervention strategies (CAIS). Thus this study did not select the severities of ADHD symptoms into logistic regression analysis to examine their association with CAIS.

Comment

In 2.1, the authors mention that the caregivers "were interviewed by research assistants by using the research questionnaire." Were there interviews in addition to questionnaires? If so, what was the reliability among the research assistants in the conduction of an interview?

Response

Thank you for your reminding. In this study research assistants explained the procedures and methods of completing the questionnaire to the participants, and then the participants completed the questionnaire by themselves. In the revised manuscript we deleted the description "the caregivers were interviewed by research assistants by using the research questionnaire." We also added the description below to illustrate the procedure of study. Please refer to line 207-212.

“Before starting the study, the PI trained the research assistants to make sure that they were competent to direct the participants to complete the research questionnaire. Then research assistants explained the procedures and methods of completing the questionnaire to the participants individually. The participants could propose any question when they had problems on completing the questionnaires, and the research assistants resolved their problems. The PI discussed with research assistants weekly to make sure the quality of the study.”

Comment

Apologies if I missed this, but what proportion of the sample received traditional ADHD treatments (i.e., medication, parent training) in the past or currently? Were any of the children still medicated and/or in parent training treatments, and if so, may need to consider in statistical models?

Response

Thank you for your reminding. Most of children without current medication treatment for ADHD have ever received medication before but stopped taking because of intolerance to adverse effects. We added the descriptions above into the revised manuscript. Please refer to line 134-140. Given that nearly all children currently received or have ever received medication for ADHD symptoms, we did not compare the level of affiliate stigma between caregivers of children with and without medication for ADHD,

“A total of 341 (85.3%) and 59 (14.7%) children received medication and group cognitive-behavioral psychotherapy currently for their ADHD symptoms. All caregivers and their children with ADHD received psychoeducation about the etiology, symptoms presentation, and treatment strategies of ADHD. Caregivers also received counseling about the skills to manage children’s ADHD symptoms and how to communicate with children. Most of children without current medication treatment for ADHD have ever received medication before but stopped taking because of intolerance to adverse effects.”

Comment

More information about the focus group process would be useful to the reader. How were caregivers selected for participation for focus groups? Were they part of the larger 400 sample? What queries were used, who led the groups, how long did groups last, etc?

Response

We added more introductions for the caregivers recruited into the focus groups as below. Please refer to line 152-156. Caregivers participating into the focus groups were not recruited into the main study.

“The family caregivers recruited into the focus groups have visited the child and adolescent psychiatric clinics for their children’s ADHD symptoms for at least two years. All participants had clear knowledge about standard treatment and CAIS for ADHD and were willing to share their experiences in the focus groups. “

We added more introductions about what queries were used, who led the groups, and how long did groups last as below. Please refer to line 156-160.

“Each group had five to eight members. The principal investigator (C-F. Yen) lead the focus groups by introducing the purpose of the focus groups and inviting the members to propose CAIS which family caregivers had employed and that the caregivers and child psychiatrists had heard of. The principal investigator also invited the members to clarify ambiguity and enhance discussion. Each focus group lasted for 40-50 minutes.”

Comment

In 2.2.1, the authors mention development of a questionnaire. It would be helpful to specify that the questionnaire was then used with the full sample of 400.

Response

The questionnaire first introduced the definitions of CAIS and then inquired whether family caregivers of children with ADHD had ever employed the 15 CAIS for managing their children’s ADHD. We added the introduction in the revised manuscript. Please refer to line 162-168.

Comment

Not all readers may be familiar with the SNAP-IV. Are there clinical cutoffs? Would be helpful to include these in the paper to ease interpretation of the means presented in Table 1. (i.e., how clinically severe are the children on average?)

Response

The SNAP-IV was used to measure the severity of ADHD and oppositional symptoms, but there was no clinical cutoff. A higher score on the subscales indicates a more severe inattention, hyperactivity/impulsivity and oppositional symptoms. The children’s scores of inattention and hyperactivity/impulsivity subscales on the SNAP-IV in the present study were 13.4 (3.6) and 9.8 (6.0), respectively. According to the norm of the Chinese Version of the SNAP- IV for ADHD in Taiwan (Liu et al., 2006), the scores indicated mild levels of severities. We added the introduction of the SNAP-IV and the interpretation in the revised manuscript. Please refer to line 189-191 and line 232-234.

Comment

In 2.3, was there a specific rationale for selecting 10% as the cutoff (as opposed to a different cutoff) for including a given strategy in logistic regressions? Response

We selected 10% as the cutoff to make sure the statistical power. We added explanation as below in the revised manuscript. Please refer to line 215-219.

“To make sure the statistical power, only the strategies employed by over 10% of the caregivers were selected into logistic regression analysis to examine the associations of affiliate stigma with the employment and effectiveness of the strategies.”

Comment

In Table 1, it might be helpful to change the font of "caregivers" and "children" to italics, bold, or underlined to help the reader quickly orient to the categories of demographics included in the table.

Response

Thank you for your suggestion. We changed them in Table 1, Table 3 and Table 4.

Comment

Tables would benefit from an indicator within the table (asterick, bolded text, or similar) to highlight which analyses are significant.

Response

We highlighted the significant results with bold letters in Table 3 and Table 4.

Comment

The sample is heavily male - is this representative of the distribution of ADHD across the sexes in Taiwan? May limit generalizability of findings to families of girls with ADHD.

Response

The children with ADHD in the present study were mainly boys (84%). A study using the National Health Insurance Research Database of Taiwan to explore trends in ADHD diagnosis from 2000 to 2011among youths in Taiwan revealed that 21.4% of cases were girls and 78.6% were boys (Wang et al., 2017). The small number of caregivers of girls in the present study may limit generalizability of findings to families of girls with ADHD. We added it as one of limitations of this study. Please refer to line 348-353.

Comment

Discussion - I wondered how perceptions of effectiveness of alternative interventions might compare to perceptions of effectiveness of traditional interventions (either in this study or more broadly in the literature). on Page 10, the authors mentioned that "treatments provided in child and adolescent psychiatric clinical units were rated by 84% of family caregivers as effective." Does this refer to traditional interventions, and was this in the current sample (I think so, but I didn't see it in the results)?

Response

We revised this sentence as below to clarify. Please refer to Results section, line 269-271 and Discussion section, line 315-317.

“Treatments, including medication and group cognitive-behavioral psychotherapy provided in child and adolescent psychiatric clinical units for ADHD were rated by 84.5% of family caregivers as effective in improving their child’s ADHD symptoms.”

Comment

Similarly, might suggest moving the Odds ratio reported in 4.2 to results section.

Response

We moved the Odds ratio reported in 4.2 to results section. Please refer to line 269-271.

Round 2

Reviewer 1 Report

I have no further comments

Author Response

We appreciate your comment and support for our manuscript.

Reviewer 2 Report

In the present version the manuscript it has been improved although the whole is weak for the practice and full of limits.

Author Response

We will continue studying in this field. We appreciate your comments on this manuscript.

Reviewer 3 Report

The authors have endeavored to respond with clarification to my original suggestions but I continue to have a few concerns that I feel were not adequately addressed:

1) I was hoping for information about potential differences in findings among various subtypes of ADHD. In their response, the authors mention "The DSM-5 has removed the subtypes for ADHD." This is not precisely correct. The DSM-5 still separates ADHD into "presentations" - inattentive, hyperactive, and combined. Thus, based on the DSM-5, it should still be possible to address my original question, which was "Regarding inclusion criteria, were all ADHD subtypes included (inattentive, hyperactive/impulsive, combined)? Would be useful to include in Table 1. Would also be interested in whether the results were impacted by subtype." If the authors do not have this information, perhaps it should be addressed as a limitation.

2) I had also pointed out the importance of considering medication use - theoretically and analytically. I appreciate that most of the children had tried medication at some point, but what percentage were currently taking medication? I think it is possible that the relationships in this study could differ between families that are currently utilizing best practice ADHD treatment (medication) and those who are not, including those who discontinued medication.

A minor point - there are grammatical concerns in the revised portions that were not evident to me in the original manuscript. I would suggest careful editing for English language conventions.

Author Response

Comment

I was hoping for information about potential differences in findings among various subtypes of ADHD. In their response, the authors mention "The DSM-5 has removed the subtypes for ADHD." This is not precisely correct. The DSM-5 still separates ADHD into "presentations" - inattentive, hyperactive, and combined. Thus, based on the DSM-5, it should still be possible to address my original question, which was "Regarding inclusion criteria, were all ADHD subtypes included (inattentive, hyperactive/impulsive, combined)? Would be useful to include in Table 1. Would also be interested in whether the results were impacted by subtype." If the authors do not have this information, perhaps it should be addressed as a limitation.

Response

Thank you for your comment. In the revised manuscript we added it as one of limitations of this study as below. Please refer to line 376-378.

“Although the present study measured children’s current ADHD symptoms, we did not determine the subgroups of children based on the DSM-5 inattentive, hyperactive, and combined presentations.”

Comment

I had also pointed out the importance of considering medication use - theoretically and analytically. I appreciate that most of the children had tried medication at some point, but what percentage were currently taking medication? I think it is possible that the relationships in this study could differ between families that are currently utilizing best practice ADHD treatment (medication) and those who are not, including those who discontinued medication.

Response

Thank you for your suggestion. We examined the differences in the application and effectiveness of CAIS and level of affiliate stigma between groups of children who received medication currently for their ADHD symptoms (n = 341) and those who received only group cognitive-behavioral psychotherapy currently (n = 59). Based on the results, we added a new paragraph in Results section as below. Please refer to line 272-285.

“3.4. Current psychiatric treatment, CAIS and affiliate stigma

Differences in the application and effectiveness of CAIS and level of affiliate stigma were compared between groups of children who received medication currently for their ADHD symptoms (n = 341) and those who received only group cognitive-behavioral psychotherapy currently (n = 59). The results indicated that there were no differences in the use of sensory integration (p = .601), exercise training (p = .782), sugar restriction (p = .974) and omega fatty acid supplementation (p = .543) between the two groups. There were no differences in the rated effectiveness of sensory integration (p = .417), sugar restriction (p = .163) and omega fatty acid supplementation (p = .441) between the two groups. However, caregivers of children who received only group cognitive-behavioral psychotherapy currently were more likely to rate exercise training as ineffective than caregivers of children who received medication currently (p = .004). Moreover, caregivers of children who received only group cognitive-behavioral psychotherapy currently had a higher level of affiliate stigma (mean = 42.4; SD = 11.8) than did caregivers of children who received medication currently (mean = 38.1; SD = 11.1; p = .006).”

We also add a new paragraph in Discussion section as below. Please refer to line 350-360.

4.3. Current psychiatric treatment, CAIS and affiliate stigma

“The present study did not find significant differences in the use and rated effectiveness of common CAIS between the children receiving medication and receiving group cognitive-behavioral psychotherapy currently. However, caregivers of children receiving psychotherapy were more likely to rate exercise training as ineffective than caregivers of children receiving medication currently. Caregivers of children receiving psychotherapy also had a higher level of affiliate stigma than did caregivers of children receiving medication currently. Tracing back the history, we found that most of children receiving psychotherapy but no medication currently had ever received medication before but stopped taking medication because of intolerance to adverse effects. It raised the possibility that caregivers’ affiliate stigma may relate to children’s intolerance to adverse effects or caregivers’ evaluation for children’s adverse response to medication. It warrants further study.”

Comment

A minor point - there are grammatical concerns in the revised portions that were not evident to me in the original manuscript. I would suggest careful editing for English language conventions.

Response

Thank you for your reminding. We reviewed the manuscript and corrected the grammatical errors.